# Impact of Carbon-Based Nanoparticles on Polyvinyl Alcohol Polarizer Features: Photonics Applications

**DOI:** 10.3390/nano14090737

**Published:** 2024-04-23

**Authors:** Natalia Kamanina, Larisa Fedorova, Svetlana Likhomanova, Yulia Zubtcova, Pavel Kuzhakov

**Affiliations:** 1Joint Stock Company Scientific and Production Corporation, S.I. Vavilov State Optical Institute, Babushkona Str. 36/1, 192171 St. Petersburg, Russia; lsv-87@bk.ru (S.L.); chozenone@mail.ru (Y.Z.); kpv_2002@mail.ru (P.K.); 2Department of Photonica, St. Petersburg Electrotechnical University (“LETI”), Ul. Prof. Popova 5, 197376 St. Petersburg, Russia; loresafyoct@gmail.com; 3Department of Advanced Development, Petersburg Nuclear Physics Institute, National Research Center “Kurchatov Institute”, 1 md. Orlova Roshcha, 188300 Gatchina, Russia; 4Department of Physics, Saint-Petersburg State University of Aerospace Instrumentation, Bolshaya Morskaia Str. 67, 190000 St. Petersburg, Russia

**Keywords:** polyvinyl alcohol thin film, graphene oxide nanoparticles, fullerenes, spectral characteristics, microhardness, wetting (contact) angle, modeling

## Abstract

Among different inorganic and organic polarizer elements, thin-film light polarizers occupy a special place because of their flexibility, ease of integration into any optoelectronic circuit, and good functioning in the visible and near-infrared spectral range and can compete with Glan and Nicolas volumetric prisms. This paper presents the results of a study on how carbon-based nanoparticles influence on the basic properties of a well-known PVA-based polymer matrix, using which it is possible to obtain good transparency for parallel light components. An accent is made on graphene oxide nanoparticles, which are used as PVA sensitizers. It was shown for the first time that the structuring of PVA with graphene oxides allows an increased transmittance of the parallel light component to be obtained, saving the transmittance of the orthogonal one. Moreover, the graphene network can increase the mechanical strength of such thin-film PVA-based polarizers and provoke a change in the wetting angle. These advantages make it possible to use graphene oxide-structured thin-film light polarizers based on a PVA matrix as an independent optoelectronic element. Some comparative results for polarizers based on PVA-C_70_ structures are shown as well.

## 1. Introduction

It is well known that polyvinyl alcohol (PVA) materials are widely used as matrix structures in general and specific technologies such as displays, transistors, microscopy, modulator and switching devices, biomedicine, acoustic optics, and other optoelectronic circuits. This is because these materials have a number of remarkable properties and unique parameters, including high transmittance in the visible and near-infrared spectral ranges, flexibility, and water solubility, as well as the possibility of changing electrical and optical characteristics.

PVA, for example, can act as a doping agent itself (with a low concentration, at about 0.08 wt.%), which can be used in order to change the conductivity and the thermal stability of some conductive polymers (such as PEDOT/PSS), as shown in [1]. At the same time, PVA can be applied to change the critical properties of the biopolymers, where the band gap and the dipole moment are important variable parameters [2]. Moreover, chitosan–polyvinyl alcohol nanocomposites were successfully tested for regenerative therapy applications [3], and PVA nanofiber as an insoluble pH sensor was shown in [4].

It should be noted that the PVA matrix is very often used as the orienting layer in the liquid crystal (LC) display and spatial light modulator (SLM) area [5]. Furthermore, PVA is a good base for creating the thin-film polarizers of light; it is flexible and operates in the UV and VIS spectral range, up to the near IR, and it can also compete with volumetric Glan prisms [6]. Based on this technology, the thin-film polarizers with improved spectral and mechanical properties, due to the structuring of a PVA matrix with carbon nanotubes (CNTs), were sensitized and shown in [7].

In modern trends, to vary the basic characteristics of a PVA matrix, there are currently effective ways to structure this matrix with nanoparticles, for example, Al_2_O_3_, TiO_2_, Au nanoparticles, carbon nanotubes, and shungites. Thus, in [8], with the introduction of the Al_2_O_3_ nanoparticles, in the concentration range from 0 to 70%, it was possible to change the dielectric parameters and the mobility of the charge carriers and to identify the variable structural modifications of this matrix when analyzing images obtained via the electron scanning microscopy. Thus, the effective use of the structured PVA compositions for the creation of the organic field-effect transistors was shown. In the publication [9], TiO_2_ nanoparticles were used for PVA doping; the concentration of the incorporated nanoparticles was varied from 0.2 to 0.4 wt.%. A change in the refractive properties, transmission, band gap width, and dielectric parameters of this sensitized polymer composite was found. The change in the luminescent properties was analyzed in the article [10] when Au nanoparticles were introduced into PVA. From the point of view of the modern achievements in the field of flexible electronics based on polymer technologies, the publication [11] is of great interest, where the polymer composition of a biodegradable poly (vinyl pyrrolidone) composition/poly (vinyl alcohol) (PVP/PVA) with SnO_2_ nanoparticles was studied. It was found that the dispersion of small amounts of the SnO_2_ nanoparticles (at the level of 1 wt.%, 3 wt.%, and 5 wt.%) in the initial matrix of the PVP/PVA mixture provides the improved thermal, optical, and dielectric properties of the obtained doped films. In [12], carbon nanotubes (CNTs) were introduced into PVA; the created composition was placed on a silicon substrate, and the temperature coefficient of the resistance (TCR) of the system was studied in the temperature range of 270−350 K. It was found that with forward displacement, the TCR has a negative sign for a pure PVA matrix and a positive sign for a carbon-nanotube-structured one. With reverse displacement, the sign of the TCR changed to the opposite. A significant change in the mechanical characteristics when introducing CNTs with a high Young’s modulus (from 0.65 to 5.5 TPa for single-walled CNTs and 0.2 to 1.0 TPa for multi-walled CNTs) into various polymer materials, including PVA, was observed and visualized by the authors of publication [13]. An earlier publication [14] revealed a 2.5-fold increase in the microhardness of the PVA films when structuring them with carbon nanotubes.

Naturally, as the class of nanoparticles expanded due to the discovery of graphene, many scientific and technical groups began to use this sensitizer. In the study presented in [15], the possibility of PVA doping with reduced graphene oxide was shown, and the refractive features and the magneto-dielectric effect were revealed. The concentration of the introduced sensitizer was varied from 0.5 to 1 wt.%. It was found that the direct band gap of the PVA and rGO–PVA composites is 4.92 eV and 3.8 eV, respectively. The results of the refractive index changing at a wavelength of 500 nm revealed the fact that the inclusion of reduced graphene oxide in a PVA polymer matrix can increase the refractive index of the composite material from 1.28 to about 2.49. In papers [16,17], PVA compounds with different graphene oxide contents (from 0.1 up to 5 wt.%) were deposited on a glass crown K8 substrate. Spectral and mechanical characteristics were studied. The best results were obtained for the PVA thin film with the content of graphene oxide close to 0.1 wt.%. An increase in the transparency for PVA with 0.1–0.3 wt.% of graphene oxide was found, and a 1.5 times increase in the microhardness was estimated, in comparison with the pure PVA matrix thin film. A recommendation for making a polarization thin film was made. The first samples of such polarizers with a concentration of graphene oxide at 0.1 wt.% with a different method of stretching the thin-film substrate were made [17], the polarizing ability was evaluated, the possible formation of a bond between the PVA molecule and graphene oxide was proposed.

Modern studies using the PVA matrix in order to optimize the features of flexible solid-state supercapacitors with high power density and rate performance, long cycle life, and high safety were shown in the paper [18]. The pressure sensor results of PVA/SiO_2_/SiC nanocomposites were shown in [19]; it was discussed that the dielectric parameters increased with the increase in pressure for various ratios of SiO_2_–SiC nanoparticles (NPs). Finally, the increase in the sensitivity was 50.18%, 52%, and 53.46% at 1 kHz, 100 kHz, and 1 MHz, respectively, when the SiO_2_–SiC NP content reached 3.6 wt.%. The fabrication and exploration of the structural, dielectric, and optical features of PVA/SnO_2_/Cr_2_O_3_ nanostructures were presented in the paper [20]. The results showed that PVA/SnO_2_/Cr_2_O_3_ nanostructures can be considered as promising nanostructures for optics, photonics, and electronics nanodevices.

Thus, the introduction of different nanoparticles makes it possible to modify the PVA matrix in an orderly manner, which leads to a change in the transmission spectra, mechanical parameters, and refractive characteristics. This, in turn, contributes to the expansion of the field of the application of PVA compounds.

In the current paper, the specific application of PVA structures doped with graphene oxides (or the fullerene C_70_) as efficient thin-film polarizers is shown. The PVA–graphene oxide polarizers are considered as the independent optoelectronic elements. It should be mentioned that in this case there is no need to use different types of substrates (e.g., glass or silicon-based) in order to place a polymer base and test its parameters. The obtained experimental results are supported by the quantum-chemical simulations.

## 2. Materials and Methods

In this study, the domestic PVA formulations of the BKK PVA 40/2 brand with a high molecular weight of 300,000 a.u. were used; these compositions are favorably distinguished from the PVA analog (No. 182480-500MG according to the Alfa Aesar and Aldrich catalog, Karlsruhe, Germany) with a molecular weight of 100,000 a.u. It should be mentioned that a film obtained from a matrix with a molecular weight of 100,000 a.u. cannot be stretched up to 3.5–3.7 times to obtain a system with a small thickness. In this case, film rupture will be manifested. Therefore, there is a limitation to the use of such composite materials in optoelectronic circuits. Compositions with a higher molecular weight (determined by the number of CH_2_ fragments and hydroxyl groups) determine the best film-forming ability of the matrix material. An 8% solution of PVA of this specified composition was prepared. Note that the production of thin films from a solution of PVA powder in water is a rather lengthy process.

First of all, the PVA powder was poured with the distilled water in the required proportion and left for 10–14 h. After the swelling of the PVA and its transition to a gel-like state, the dissolution process began. To obtain a homogeneous transparent solution in water, a vessel with the swollen PVA was placed in a water bath, and an agitator was additionally turned on for constant stirring. The temperature of the water bath was ~100 °C. The stirring device should make no more than 1 revolution per second. If this requirement is not met and the agitator speed is increased, air bubbles will form in the PVA solution, which will lead to heterogeneity of the cast film. The process of dissolving PVA in a water bath with constant stirring took about 3–4 h. Two separate 8% PVA solutions were prepared for further sensitization with carbon nanoparticles. For the first, a domestic 1% aqueous solution of graphene oxide (obtained from Nano Tech Center, Tambov, Russia) was used as a sensitizer. It should be noted that the method of sensitizing graphene oxide NPs is written in [21]. The variation in the graphene oxide (GrO) sensitizer in the PVA matrix solution was in the range from 0.05 to 0.3 wt.%. For the second, fullerene C_70_ powder (Alfa Aesar Co., Karlsruhe, Germany, 97% C_70_, #10115890) in a concentration of 0.1 wt.% was applied for doping the PVA. The PVA mixture was placed and formed on a K8 crown glass substrate. After drying for a day, the films were stretched in a stretching machine with different stretches, ~1.7 and ~3.5 times. Further, the polarized films were tested by various methods.

To study the spectral characteristics, an SF-26 spectrophotometer was used, operating in the wavelength range of 200–1200 nm. Calibrated filters were used to control the spectral measurements in the visible spectral range. The error in the measurements of the spectra was about 0.2%. The spectra in the IR range were diagnosed on an FSM-1211 device (“Nica-Garant+”, Saint-Petersburg, Russia). This was done in order to find the most suitable PVA matrix sensitizers in the near future, which would allow our polarizers to operate in the near- and medium-IR spectral range. The microhardness values were tested on a PMT-3M device (LOMO Co., St. Petersburg, Russia). The wetting angle was measured by use of an OCA 15EC instrument, acquired from LabTech Co. (Saint Petersburg-Moscow, Russia). Quantum-chemical modeling was performed using the programs GaussView5.0 and Gaussian 09W [22,23,24]; the calculation method and the Hartree–Fock atomic basis set HF/STO-3G SP were applied.

## 3. Results

The general view of the developed thin-film polarizer is shown in Figure 1. The thickness of the sensitized PVA thin film was 100 microns when stretching 1.7–1.8 times was used. The thickness decreased to 65–80 microns when the stretching was 3.5–3.7 times.

It can be seen that, in general, the film turns out to be heterogeneous. But, upon further investigation or application, the central part of the 40 × 40 mm size was cut out. In this case, the inhomogeneities were partially avoided.

It can be mentioned that the origin of the polarizing parameters of the PVA films is possibly explained by the influence of the orientation of the graphene oxide links. Graphene oxide is a derivative of graphene with active functional groups (carboxylic, carbonyl, and hydroxyl). Due to the Van der Waals forces of the attraction between the graphene planes, a self-assembling structure is possible. Such a carbon structure was considered as a framework along which additional orientation of the molecular targets of the PVA would occur. Moreover, the stretching in one direction in a stretching machine, which varies from ~1.7 to 3.5–3.7 times, can be responsible for the orientation of the PVA lamellas in the preferable direction.

The basic experimental results are shown in Figure 2 and Figure 3 and presented in Table 1 and Table 2. One can see from Figure 2 that the introduction of the NPs in the PVA matrix can provoke an increase in the transmittance of the parallel light component, saving the little transmittance for the orthogonal one.

Analyzing the data shown in Figure 2, one can testify that the best transmittance for the parallel light components is obtained for the PVA matrix doped with 0.1 wt.% of graphene oxide and fullerene C_70_ nanoparticles with the same content. The value of the transmittance obtained for these parallel components in the spectral range of 450–900 nm is close to 75% and more. The orthogonal component transmittance is close to 0.1–5% in the spectral range of 500–750 nm. Thus, the polarization ability (degree of polarization) is very high and reaches up to 100% in the visible range of the spectrum. The calculation of the degree of the polarization was performed according to Equation (1) accepted from the books [25,26]:
(1)
$$P=\frac{I_{par}-I_{orth}}{I_{par}+I_{orth}},$$

where *I*_par_ is the transmission of the parallel component of the light beam; *I*_orth_ is the transmission of the orthogonal component of the light beam.

Thus, the degree of polarization for the PVA + graphene oxide thin film is changed from 80% to 60% in the spectral range of 400–800 nm. The maximum values are close to 99% for the wavelength of 550–700 nm.

It should be noted that some comparisons can be made to check the transmittance of our polarizer and commercial polarizers based on an organic matrix. Thus, polarizers based on organic materials, such as XP38, XP40HT, XP42, and XP42HE, produced by Edmund Optics Manufacturer, Barrington, NJ, USA [27], reveal a thickness of 180 microns, working diameter of 3 mm, and an operating spectral range of 400–750 nm, and the transmission of the parallel component is over 30%. One can see that the polarizers shown in the current study have less thickness and higher transmittance of the parallel light component. This allows us to use the polarizers we are developing in any optoelectronic circuits in order to preserve their design and small dimensions while maintaining high transmission.

At the same time, a grid of carbon nanoparticles is a guide that forms a regular stacking of PVA molecules, which are subjected to varying degrees of stretching. Some results on how the carbon grid can influence the stretching process are presented in Figure 3.

The grid of the carbon NPs can predict better mechanical and wetting phenomena as well.

The data of the microhardness parameters for the PVA structured with 0.1 wt.% GrO are presented in Table 1.

Looking at the data in Table 1, it can be seen that increasing the stretching of the sensitized PVA film is increasingly influenced by the formed graphene grid. The same tendency was found with other loads that were applied to the sensitized PVA film with varying degrees of stretching. Indeed, the mechanical parameters have become better for the PVA film with the content of the graphene oxide of 0.15 wt.%, but the spectral parameters, which are important for use in polarization optics, in the optoelectronic set-up with the specific design, when it comes to miniaturizing devices, are more acceptable for the PVA thin film with 0.1 wt.% of graphene oxide.

It should be mentioned that the data presented in Table 1 are in good agreement with the effect of the mechanical parameter enhancement shown in the paper [27].

The data of the wetting angle change are shown in Table 2. Early obtained results for the PVA structured with the fullerene C_60_ and with the CNTs are presented as well. It should be mentioned that the graphene oxide-sensitized PVA compositions were also used here with a degree of stretching at the level of 1.8 times. This permits the dimension of the sensitized PVA thin-film polarizers to be established close to the mobile phone size.

**Table 2 nanomaterials-14-00737-t002:** The data of the obtained wetting (contact) angle for PVA sensitized with graphene oxide, fullerene C_60_, C_70_, and CNTs.

Content of the NPs	Wetting Angle, Degree	References
Pure PVA film	40	[28]
PVA + 0.1 wt.% C_60_	83	[28]
PVA + 1 wt.% CNTs	82	[28]
PVA + 0.05 wt.% GrO	33–38	current
PVA + 0.1 wt.% GrO	75–76	current
PVA + 0.15 wt.% GrO	106	current
PVA + 0.3 wt.% GrO	65–66	current
PVA + 0.1 wt.% C_70_	63–65	current

It should be noted that with a lower concentration of graphene oxide (0.05 wt.%) in the PVA, the carbon mesh most likely does not have time to form in order to significantly affect the strength of this matrix. With an increase in the concentration of the introduced sensitizer (0.1 and 0.15 wt.%), such a grid, due to a durable carbon frame, significantly changes the surface structure of the PVA. Moreover, the PVA molecule contains hydroxyl groups that can form aqueous bonds with the oxygen ion of the graphene functional groups. This contributes to the formation of a more ordered structure with good spectral characteristics and acceptable wetting parameters. At the highest concentration of graphene oxide (0.3 wt.%), most likely, a violation of the surface layer occurs with the rearrangement of the polymer chains, which leads to a decrease in the wetting angle. For this content of the sensitizers, poor transmittance of the parallel light component was found (please see data from Figure 2). In order to find a compromise between the spectral parameters and the wetting angle features, the graphene oxide content of 0.1 wt.% should be chosen.

It should be mentioned that the data obtained for the contact angle in the current research are correlated with the fact established in the paper [29] when bio-based polymers were doped with graphene nanoplatelets. It was discussed that contact angle measurements provide information about surface properties, indicating a surface with hydrophobic characteristics for the studied materials. Moreover, in this study, it was established that an increase in the graphene particle content has a major effect on the hardness, tensile and flexural moduli, elongation at break, and toughness. But we should note that there is no clear linear relationship between an increase in the concentration of the carbon nanoparticles and an increase in mechanical parameters of the bio-based polymers in this article. For example, the tensile strength is larger for the structured polymer with 1.0 wt.% of graphene nanoplatelets, in comparison with the materials doped with 0.5 wt.% of the nanoparticles. However, the impact strength of the doped polymer is larger for the materials with 0.5 wt.% of the dopants, in comparison with the material doped with 1.0 wt.% of the graphene sensitizers. Furthermore, a dramatic change in mechanical property improvement of resin nanocomposites with graphene oxide was shown in the paper [30] at the content of the graphene oxide in the range of 0.5–1.5 wt.%.

Our results shown in Table 2 are correlated with this fact shown in the papers [29,30] for other polymer materials structured with graphene particles as well.

It should be noted that thin-film polarizers can be made from inorganic materials as well, as was shown in [31,32,33,34,35], but it is very complicated to grow inorganic materials based on lithium niobate without defects. Moreover, these types of polarizers are not flexible, which can eliminate their application in complex optoelectronic schemes. Furthermore, polarized light can be obtained by the use of metasurfaces [36], but this is a complicated procedure as well.

It should be mentioned that the quantum-chemical simulation data presented in Figure 4 can support the obtained experimental ones with a good advantage.

In this model (please see Figure 4), 238 carbon atoms, 43 oxygen atoms, and 116 hydrogen atoms were used, including 2 carbon atoms, 4 hydrogen atoms, and 1 oxygen atom for one PVA fragment; thus, using three PVA fragments, 6 carbon atoms, 12 hydrogen atoms, and 3 oxygen atoms were taken into account. The structure of the GrO contains 232 carbon atoms, 104 hydrogen atoms, and 40 oxygen atoms. One can see that the graphene oxide can really form the network, which permits the PVA molecules to be oriented with a good advantage.

Of course, the synthesis and investigation of the properties of PVA with other carbon-containing nanoparticles, for example, with shungite, will be worthwhile in the future. It should be mentioned that shungite nanoparticles with a content of 0.1 wt.% can be considered as the possible effective dopant of a PVA base. Shungite is a natural material representing a multilevel fractal structure of graphene fragments (1 nm), as well as secondary and tertiary levels of stacks (1.5–2.5 nm) and globular stack compositions with dimensions of 6 nm; i.e., the structure of shungite can be represented as a multistage fractal grid of sheets of reduced graphene oxide [37,38,39].

Moreover, it should be mentioned that variations in the nature of carbon nanosensitizers permit the spectral range of PVA-based polymer compositions to be varied; thus, they can provoke the operation of doped polarizers in the near- and middle-IR range.

## 4. Conclusions

To summarize the results, it should be mentioned that the structuring of a PVA matrix with graphene oxide is a prospective way to optimize the spectral, mechanical, and wetting features in order to use these types of polarizers in different photonics applications. Comparative results for PVA-C_70_ polarizing thin films are presented for comparison.

From the optical point of view, it is very important to obtain the thin-film polarizers with good polarization ability for the visible and near-infrared spectral range. These thin-film polarizers can replace volumetric Glan prisms, which are classically used in optoelectronic schemes. But, via application of the thin-film polarizing construction, one can improve the design of the set-up and can carry out the miniaturization of the scheme with a good advantage.

From the technological point of view, it is a simpler procedure to sensitize thin-film polarizers based on a polymer PVA matrix in comparison to the complicated procedure connected with crystal growth and defect elimination for volumetric inorganic polarizers.

From the commercial point of view, thin-film polarizers of light can be used in any optoelectronic circuits where it is necessary to keep the dimensions of the devices small and to be able to simply glue these thin-film polarizers to the input and output substrate, for example, a liquid crystal (LC) cell or a light modulator based on an electro-optical LC mesophase.

From the point of view of the use of thin-film polarizers in a humid atmosphere, the use of graphene oxide increases the mechanical parameters such as the microhardness and contact angle of water molecules on the surface of the material presented in the current study.

Finding a compromise between good spectral parameters and acceptable mechanical and humidity characteristics, the authors believe that the concentration of the injected dopant on the base of graphene oxide is best chosen at the level of 0.1 wt.%, taking into account the PVA material as the matrix.

Together with the analytical calculation of polarization ability, quantum-chemical simulation can realize and support the influence of the graphene network on the PVA molecule orientation.

The results of this study can be involved in the education process as well because the technological process is well visualized. To obtain this polarization thin film, students can check their phones for which technologies were used in their creation: LC or LED. In the case of an LC matrix, then when you turn the polarizer, you can see the illumination or dimming of the phone screen. This is of extreme interest to young researchers. Thus, these measurements and observations will stimulate the collection of knowledge of young scientists in the material science area with a good advantage.

## Figures and Tables

**Figure 1 nanomaterials-14-00737-f001:**
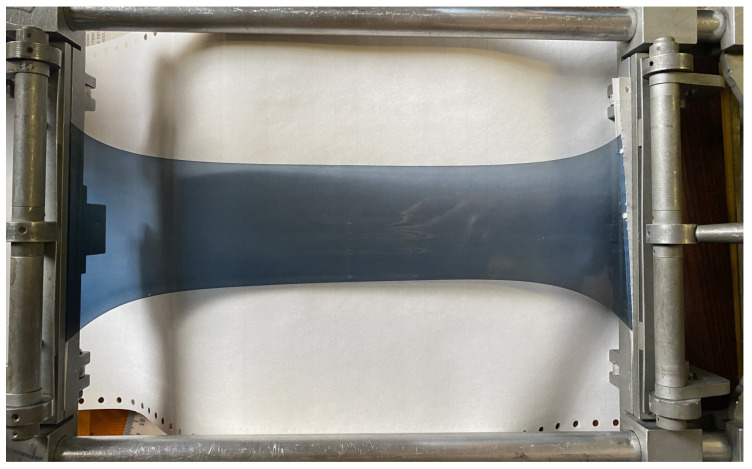
General view of the synthesized polarizing film in a stretchable frame.

**Figure 2 nanomaterials-14-00737-f002:**
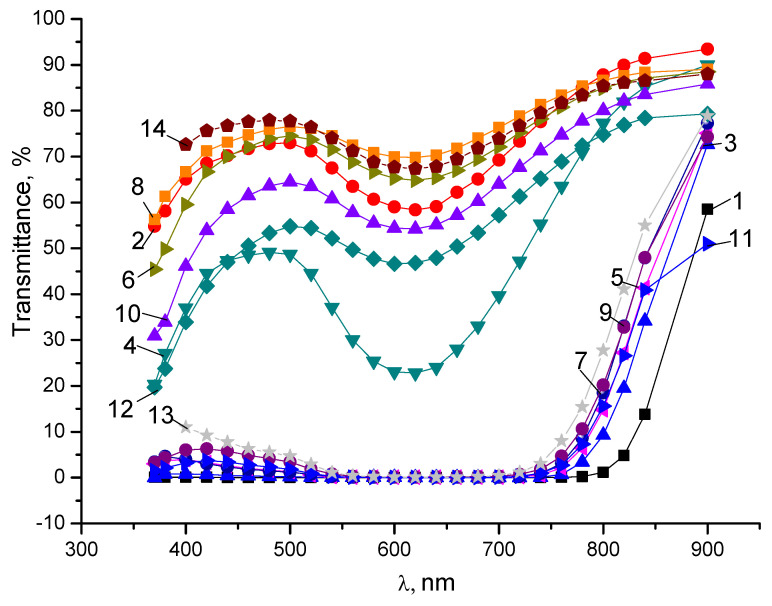
The transmittance of the parallel (*T* par.) and orthogonal (*T* orth.) light components for the thin-film polarizers with the stretching of 3.5 times (long) and of 1.8 times (short). Curves are the following: 1—*T* orth. pure PVA, long; 2—*T* par. pure PVA, long; 3—*T* orth. pure PVA, short; 4—*T* par. pure PVA, short; 5—*T* orth. PVA + 0.05 wt.% GrO, short; 6—*T* par. PVA + 0.05 wt.% GrO, short; 7—*T* orth. PVA + 0.1 wt.% GrO, short; 8—*T* par. PVA + 0.1 wt.% GrO, short; 9—*T* orth. PVA + 0.15 wt.% GrO, short; 10—*T* par. PVA + 0.15 wt.% GrO, short; 11—*T* orth. PVA + 0.3 wt.% GrO, short; 12—*T* par. PVA + 0.3 wt.% GrO, short; 14—*T* par. PVA + 0.1 wt.% C_70_ short; 13—*T* orth. PVA + 0.1 wt.% C_70_ short.

**Figure 3 nanomaterials-14-00737-f003:**
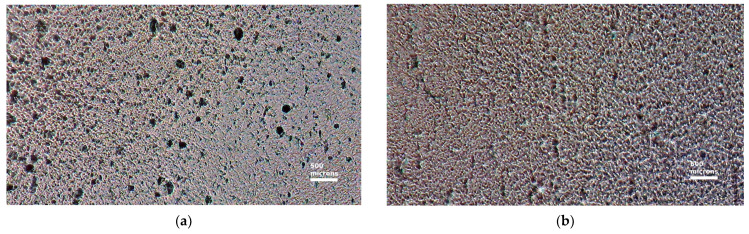
Micrographs showing the state of the PVA films with 0.1 wt.% graphene oxide before stretching (**a**) and after stretching (**b**) in a stretching machine.

**Figure 4 nanomaterials-14-00737-f004:**
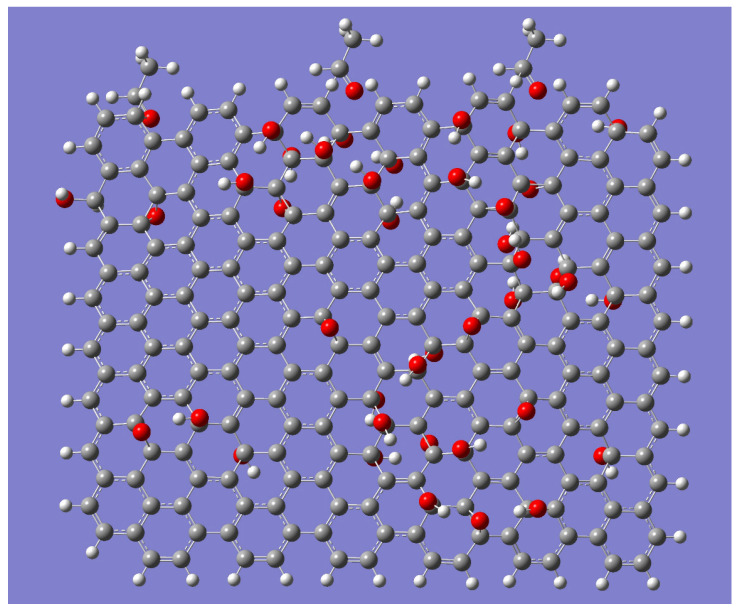
Quantum-chemical modeling of the PVA matrix interaction with the graphene oxide structures. The carbon atoms are collared with grey; oxygen atoms—with red; hydrogen atoms—with white.

**Table 1 nanomaterials-14-00737-t001:** Microhardness of the 8% PVA matrix structure doped with the different contents of the GrO sensitizer. The load was 2 g; the stretching force was 1.8 times and 2.5 times.

Microhardness, MPa				
1.8 times	0.05 wt.% GrO	0.1 wt.% GrO	0.15 wt.% GrO	0.3 wt.% GrO
	30.20	20.50	35.18	32.41
2.5 times	22.81	36.19	48.59	37.07

## Data Availability

Publicly available data on classical preparation of PVA-based polarizers with CNTs can be found in the patent: “Polarizing films for the visible range of the spectrum with a nanostructured surface based on carbon nanotubes”, Patent for invention No. 2426157 (RU 2426157 C1), Priority dated 9 March 2010; registered in the State Register of Inventions of the Russian Federation on 10 August 2011. Authors: N.V.Kamanina, P.Ya. Vasilyev, V.I. Studenov.

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
