# Peer review of "Impact of Carbon-Based Nanoparticles on Polyvinyl Alcohol Polarizer Features: Photonics Applications"

_nanomaterials, 2024, doi:10.3390/nano14090737_

Round 1

Reviewer 1 Report

Comments and Suggestions for Authors

In this paper, the PVA structure doped with the graphene oxides is shown as the efficient thin-film polarizer. The research results presented by the authors are interesting. But the organization and writing of the manuscript need to be improved. Major revisions are recommended for acceptance.

1.     The thickness, width and length of the polarizing film in Fig.1 should be given.

2. The thin film in Figure 1 appears to have significant non-uniformity. How does this non-uniformity occur and what impact does it have on the optical performance of polarizer?

3. The author should compare the performance of the polarizers proposed in this article with other typical polarizers that currently have excellent performance. 

4. It is better to give more detailed information of the devices used in the experiment. 

5. What's the advantages of the proposed thin-film PVA-based polarizers? It is better to emphasize the advantages of the proposed thin-film.

6. The language in this paper must be improved before it can be accepted for publication.

Comments on the Quality of English Language

The language in this paper must be improved before it can be accepted for publication.

Author Response

Dear Reviewer!

Thank you for your remarks and recommendation!

I have included the answers to your recommendation in the text body.

All paragraphs added are collared with yellow.

Thanks a lot once again!

Best Regards,

Natalia Kamanina

Comments and Suggestions for Authors

In this paper, the PVA structure doped with the graphene oxides is shown as the efficient thin-film polarizer. The research results presented by the authors are interesting. But the organization and writing of the manuscript need to be improved. Major revisions are recommended for acceptance.

  1. The thickness, width and length of the polarizing film in Fig.1 should be given. Thank you! The additional paragraph has been added. Please see the information below.

The general view of the developed thin-film polarizer is shown in Figure 1. The thickness of the sensitized PVA thin-film was 100 microns, if stretching 1.7-1.8 times has been used. The thickness decreased to 65-80 microns if the stretching was 3.5-3.7 times.

  1. The thin film in Figure 1 appears to have significant non-uniformity. How does this non-uniformity occur and what impact does it have on the optical performance of polarizer?

Yes. Thank you for an attentive approach! A paragraph has been added after Figure 1.

It can be seen that, in general, the film turns out to be heterogeneous. But, upon further investigation or application, the central part of the 40´40 mm size was cut out. In this case, the inhomogeneities were partially avoided.

  1. The author should compare the performance of the polarizers proposed in this article with other typical polarizers that currently have excellent performance. 

Yes, it is very good recommendation. Sorry, at the present time some cites are forbidden for the Russian scientists. But, I can consider the performance of other polarizers based on the organic matrix, for example, the paragraph has been added after Fig.1 as well:

It should be noticed that some comparison can be made to check the transmittance of our and commercial polarizers based on organic matrix. Thus, polarizers based on organic materials, such as: XP38, XP40HT, XP42, XP42HE, produced by Edmund Optics Manufacturer, USA, reveal the thickness from 180 microns, working diameter from 3 mm and the operating spectral range of 400-750 nm, the transmission of the parallel component is over 30%. One can see that polarizers shown in the current study has less thickness and higher transmittance of the parallel light component. This allows us to use the polarizers we are developing in any optoelectronic circuits in order to preserve their design and small dimensions, while maintaining high transmission.

  1. It is better to give more detailed information of the devices used in the experiment.

Sorry, maybe I have not understood you correctly. The information about the devices used to establish the thin-film polarizers features have been shown in section  “Materials and methods”, last paragraph..

  1. What's the advantages of the proposed thin-film PVA-based polarizers? It is better to emphasize the advantages of the proposed thin-film.

Thank you! I have added the paragraph in the Conclusion section.

From the commercial point of view, the use of thin-film polarizers of the light allows them to be used in any optoelectronic circuits where it is necessary to keep the small dimensions of the devices, as well as to be able to simply glue these thin-film polarizers to the input and output substrate, for example, an liquid crystal (LC) cell or a light modulator based on an electro-optical LC-mesophase.

  1. The language in this paper must be improved before it can be accepted for publication. Comments on the Quality of English Language

The language in this paper must be improved before it can be accepted for publication.

Thank you! I have seen the text and improve some sentences and include articles.

Reviewer 2 Report

Comments and Suggestions for Authors

The authors have characterized the polarizing performance of stretched PVA films doped with graphene oxides. The manuscript need to response to the following comments before being accepted.

(1)     The introduction part is too tedious and not to the point. The authors should summarize the progress of the performance of PVA films doped with graphene oxides.

(2)     The physical origin of the polarization performance of the film should be explained.

(3)     The influence of the molecular weight of the PVA powder on the polarizing performance should be analyzed and discussed.

(4)     The parallel transmittance of the polarizer was around 70%. The working parameter should be discussed in terms of the commercial PVA polarizer doped with Iodine.

(5)     The presentation of the experimental data should be optimized. For example, the curves in Fig. 2 are interlaced and it is hard for readers to discriminate and understand.

(6)     The English is difficult to understand.

Comments on the Quality of English Language

Need to be improved.

Author Response

Dear Reviewer!

Thank you for your remarks and recommendation!

I have included the answers to your recommendation in the text body.

All paragraphs added are collared with green and yellow.

Thanks a lot once again!

Best Regards,

Natalia Kamanina

Comments and Suggestions for Authors

The authors have characterized the polarizing performance of stretched PVA films doped with graphene oxides. The manuscript need to response to the following comments before being accepted.

  • The introduction part is too tedious and not to the point. The authors should summarize the progress of the performance of PVA films doped with graphene oxides.

I'm sorry, maybe I didn't quite understand your recommendation exactly. I believe that the text of the introduction tells about the problem of optimizing the PVA matrix, as well as about improving its properties during doping.

  • The physical origin of the polarization performance of the film should be explained.

Thank you! Nice question-recommendation! I have added some paragraph after Fig.1.

It can be mentioned that the origin of the polarizing parameters of the PVA films is possibly explained by the influence of the orientation of the graphene oxide links.  Really, the graphene oxide is a derivative of the graphene with active functional groups (carboxylic, carbonyl and hydroxyl). Due to the Van der Waals forces of the attraction between the graphene planes, a self-assembling structure is possible. Such a carbon structure was considered as a framework along which additional orientation of the molecular targets of the PVA would occur. Moreover, the stretching in one direction in a stretching machine, which varies from 1.7 to 3.7 times can be responsible for the orientation of the PVA lamellas in the preferable direction.

  • The influence of the molecular weight of the PVA powder on the polarizing performance should be analyzed and discussed.

Yes, it is true. We have some experience to check the influence of that molecular weight of 100 000 and 300 000 a.u. The film obtained from a matrix with a molecular weight of 100,000 AU does not allow it to be stretched up to 3.5 times to obtain a system with a small thickness. A film rupture is manifested. Therefore, there is a limitation to use such composite materials in optoelectronic circuits, where it is required to maintain small dimensions or simply paste polarizers on functional elements, say, liquid crystal (LC)  cells.

I have modified the first paragraph in the section “Materials and methods”

In this study, the domestic PVA formulations of the BKK PVA 40/2 brand with a high molecular weight of 300,000 a.u. were used, which favorably distinguishes these compositions from the PVA analog (No. 182480-500MG according to the Alfa Aesar and Aldrich catalog) with a molecular weight of 100,000 a.u. It should be mentioned that the film obtained from a matrix with a molecular weight of 100,000 a.u. does not allow it to be stretched up to 3.5-3.7 times to obtain a system with a small thickness. A film rupture is manifested. Therefore, there is a limitation to use such composite materials in optoelectronic circuits. The compositions with a higher molecular weight (determined by the number of CH2 fragments and hydroxyl groups) determine the best film-forming ability of the matrix material. A 8% solution of the PVA of this specified composition was prepared. Note that the production of the thin films from a solution of the PVA powder in water is a rather lengthy process.

  • The parallel transmittance of the polarizer was around 70%. The working parameter should be discussed in terms of the commercial PVA polarizer doped with Iodine.

Yes. It is true! Good remark! I have added the following paragraph in the section “Results” after Fig.2.

It should be noticed that some comparison can be made to check the transmittance of our and commercial polarizers based on organic matrix. Thus, polarizers based on organic materials, such as: XP38, XP40HT, XP42, XP42HE, produced by Edmund Optics Manufacturer, USA [27], reveal the thickness from 180 microns, working diameter from 3 mm and the operating spectral range of 400-750 nm, the transmission of the parallel component is over 30%. One can see that polarizers shown in the current study has less thickness and higher transmittance of the parallel light component. This allows us to use the polarizers we are developing in any optoelectronic circuits in order to preserve their design and small dimensions, while maintaining high transmission.

  • The presentation of the experimental data should be optimized. For example, the curves in Fig. 2 are interlaced and it is hard for readers to discriminate and understand.

I'm sorry, maybe I didn't quite understand your recommendation exactly. I believe that the Fig.2 and the captions to Figure 2 can explain the obtained feature with good advantage.

(6)     The English is difficult to understand.

Comments on the Quality of English Language.

OK. Thank you! I have seen the text body and modify a little bit the sentences and articles.

Reviewer 3 Report

Comments and Suggestions for Authors

The manuscript is quite interesting and it has been written in a clear way.

Anyway I have a few minor comments that need to be addressed to improve the quality of the paper.

Line 270:

can supported --- > can support 

Lines 317-320:

Actually, this last sentence is not very clear. I don't understand in which way this research can help the education process. Please clarify or eliminate it.

Author Response

Dear Reviewer!

Thank you for your delicate remarks and recommendation!

I have included the answers to your recommendation in the text body.

All paragraphs added are collared with blue.

As an additional, some paragraphs have been collared with yellow and green.

Thanks a lot once again!

Best Regards,

Natalia Kamanina

Comments and Suggestions for Authors

The manuscript is quite interesting and it has been written in a clear way.

Anyway I have a few minor comments that need to be addressed to improve the quality of the paper.

1). Line 270:

can supported --- > can support 

Thank you! I have corrected this one according to your remark.

It should be mentioned that the quantum-chemical simulation data presented in Figure 4 can support the obtained experimental ones with good advantage. 

2). Lines 317-320:

Actually, this last sentence is not very clear. I don't understand in which way this research can help the education process. Please clarify or eliminate it.

Thank you very much! Nice recommendation! I have modified the paragraph:

The results of this study can be involved in the education process as well due to the reason, that the technological process is good visualized. Really, to obtain this polarization thin-film the students can check their phones for which technologies were used in their creation: LC or LED. If this is an LC-matrix, then when you turn the polarizer, you can see the illumination or dimming of the phone screen. This is of extreme interest to young researchers. Thus, these measurements and observation will stimulate the collection of the knowledge of the young scientists in the material science area with good advantage.

Round 2

Reviewer 1 Report

Comments and Suggestions for Authors

In Section "2. Materials and Methods", the author mentioned "The spectrum in the IR range was diagnosed on the FSM-1211 device (Infraspec Co.)". Why and how was the FSM-1211 used for such diagnosis? The author should provide more detailed description. 

Comments on the Quality of English Language

The quality of English language need to be improved.

Author Response

Dear Reviewer!

Thank you for your kind and useful consideration of the paper.

I have seen you second round comments and tried to answer to them.

All added corrections are collared with pink.

Best Regards,

Natalia Kamanina

Comments and Suggestions for Authors

In Section "2. Materials and Methods", the author mentioned "The spectrum in the IR range was diagnosed on the FSM-1211 device (Infraspec Co.)". Why and how was the FSM-1211 used for such diagnosis? The author should provide more detailed description. 

Wow, how carefully you reacted to the recommendations of the article. Wonderful! Thank you! I have added a little sentence. For, example, it can be possible to use our polarizers for the manipulations with light at the wavelength at 805 nm (titanium sapphire lasers), at 1315 nm (iodine laser), etc. It will be considered in the separated future publications.

The spectra in the IR range were diagnosed on the FSM-1211 device (Infraspec Co.). This was done in order to find the most suitable PVA matrix sensitizers in the near future, which would allow our polarizers to operate in the near and medium IR spectral range.

Comments on the Quality of English Language. The quality of English language need to be improved.

Well. I am trying to improve English. Indeed, now we have nor realized wide our participations at the international conferences, the English language is not improving for this reason. I have collared the corrections by pink as well.

Thank you very much once again! You are good reviewer! Your remarks permit to dramatically improve the paper text for the understanding.

Best Regards,

Natalia Kamanina

=======================================

Natalia V. Kamanina (Prof., Dr.Sci., PhD)

Head of the lab for Photophysics of media with nanoobjects

Vavilov State Optical Institute

Kadetskaya Liniya V.O., dom.5, korpus 2,

St.- Petersburg, 199053, Russia

Professor of the St.-Petersburg Electrotechnical University (“LETI”),

Part-time Leading Researcher at Nuclear Physics Institute (Gatchina)

Job phone: +7 (812) 327-00-95

Fax: +7 (812) 331-75-58 (for N.V.Kamanina)

Lab_cite: sites.google.com/view/photophysics-lab

https://publons.com/researcher/1696479/natalia-kamanina/

https://sciprofiles.com/news-feed

http://rusnor.org/network/webinars/10203.htm

http://www.npkgoi.ru/?module=articles&c=profil&b=7

http://www.nanometer.ru/2007/08/09/liquid_crystal_3905.html

https://etu.ru/ru/fakultety/fakultet-elektroniki/sostav-fakulteta/kafedra-fotoniki/sostav-kafedry

=======================================

Reviewer 2 Report

Comments and Suggestions for Authors

The authors have addressed all my conerns

Author Response

Dear Reviewer!

Thank you for your kind and useful consideration of the paper.

I have seen you second round comments. O!!!  Thank you once again!

Your recommendations permit to improve the paper with good advantage.

A little correction made by me now I have collared with pink.

Best Regards,

Natalia Kamanina

Comments and Suggestions for Authors

The authors have addressed all my conerns

=======================================

Natalia V. Kamanina (Prof., Dr.Sci., PhD)

Head of the lab for Photophysics of media with nanoobjects

Vavilov State Optical Institute

Kadetskaya Liniya V.O., dom.5, korpus 2,

St.- Petersburg, 199053, Russia

Professor of the St.-Petersburg Electrotechnical University (“LETI”),

Part-time Leading Researcher at Nuclear Physics Institute (Gatchina)

Job phone: +7 (812) 327-00-95

Fax: +7 (812) 331-75-58 (for N.V.Kamanina)

Lab_cite: sites.google.com/view/photophysics-lab

https://publons.com/researcher/1696479/natalia-kamanina/

https://sciprofiles.com/news-feed

http://rusnor.org/network/webinars/10203.htm

http://www.npkgoi.ru/?module=articles&c=profil&b=7

http://www.nanometer.ru/2007/08/09/liquid_crystal_3905.html

https://etu.ru/ru/fakultety/fakultet-elektroniki/sostav-fakulteta/kafedra-fotoniki/sostav-kafedry

=======================================
